# Mechanisms and Virulence Factors of *Cryptococcus neoformans* Dissemination to the Central Nervous System

**DOI:** 10.3390/jof10080586

**Published:** 2024-08-17

**Authors:** Ammar Mutahar Al-Huthaifi, Bakeel A. Radman, Abdullah Ali Al-Alawi, Fawad Mahmood, Tong-Bao Liu

**Affiliations:** 1Medical Research Institute, Southwest University, Chongqing 400715, China; ammaralhuthaifi@gmail.com (A.M.A.-H.); alalawisky@tu.edu.ye (A.A.A.-A.); fawadsawabi123@gmail.com (F.M.); 2Cancer Center, Integrated Hospital of Traditional Chinese Medicine, Southern Medical University, Guangzhou 510315, China; bakeelali55@gmail.com; 3Jinfeng Laboratory, Chongqing 401329, China; 4Engineering Research Center for Cancer Biomedical and Translational Medicine, Southwest University, Chongqing 400715, China

**Keywords:** cryptococcosis, blood–brain barrier, respiratory tract infection, central nervous system infection, *Cryptococcus neoformans*, Trojan horse

## Abstract

Cryptococcosis is a prevalent fungal infection of the central nervous system (CNS) caused by *Cryptococcus neoformans*, a yeast with a polysaccharide capsule in the basidiomycete group. Normally, *C. neoformans* infects the respiratory tract and then breaches the blood–brain barrier (BBB), leading to meningitis or meningoencephalitis, which leads to hundreds of thousands of deaths each year. Although the mechanism by which *C. neoformans* infiltrates the BBB to invade the brain has yet to be fully understood, research has revealed that *C. neoformans* can cross the BBB using transcellular penetration, paracellular traversal, and infected phagocytes (the “Trojan horse” mechanism). The secretion of multiple virulence factors by *C. neoformans* is crucial in facilitating the spread of infection after breaching the BBB and causing brain infections. Extensive research has shown that various virulence factors play a significant role in the dissemination of infection beyond the lungs. This review explores the mechanisms of *C. neoformans* entering the CNS and explains how it bypasses the BBB. Additionally, it aims to understand the interplay between the regulatory mechanisms and virulence factors of *C. neoformans*.

## 1. Introduction

*Cryptococcus neoformans* was initially discovered and identified as a human pathogen in the 1890s [1]. According to the World Health Organization, *C. neoformans* has been classified as a critically prioritized pathogen responsible for approximately 112,000 annual deaths, including 19% of HIV-associated deaths [2]. Globally, there are approximately 250,000 reported cases of cryptococcal meningitis each year, leading to an estimated 181,000 deaths [3]. Without treatment, the mortality rate for this infection reaches 100% [3]. *C. neoformans* is a significant fungal pathogen that significantly affects the human central nervous system, causing substantial morbidity and mortality [4]. It can potentially cause pneumonia and meningitis [5,6,7,8]. Cryptococcal meningitis is responsible for 15% of acquired immunodeficiency syndrome (AIDS)-related deaths and is more prevalent among individuals living with Human Immunodeficiency Virus (HIV) [9,10].

The two primary pathogenic types that can cause diseases of the neurological system are *C. neoformans* and *C. gattii*. *C. neoformans* usually affects people with weakened immune systems or HIV, while *C. gattii* often infects healthy individuals [4,11,12]. Some people affected by *C. neoformans* infection may show symptoms of nervous system disease [13]. These individuals typically have compromised immune systems and may have conditions such as AIDS [14,15]. After the initial lung infection, the fungus spreads to lung tissue, enters the bloodstream, and can breach the blood–brain barrier (BBB), making it the leading cause of cryptococcal meningitis [16]. This condition primarily affects the neurological system but also infects other parts of the body, such as the skin, bones, eyes, prostate, and genitourinary tract [17]. The central nervous system has three main barriers: the BBB, the CSF–brain barrier, and the blood–cerebrospinal fluid (BCSFB) barrier. Among these barriers, the BBB has received the most extensive research attention. However, all three barriers share functional characteristics aimed at safeguarding the brain against toxins, maintaining brain cell metabolism, and preventing the invasion of pathogens [18,19].

The BBB is a selectively permeable barrier separating the brain tissue from the blood vessels [20]. It comprises endothelial cells, pericytes, and astrocytes [21]. The inner lining of the brain vasculature consists of endothelial cells, which are interconnected by tight junctions. During inflammation, circulating leukocytes can enter the brain parenchyma through the endothelial layer via various mechanisms: paracellularly, by passing through tight junctions, or transcellularly, through endocytosis facilitated by the endothelial cells themselves, or by taking advantage of phagocytes as a “Trojan horse” [22,23]. The attachment and internalization of *C. neoformans* rely on transmembrane proteins, including hyaluronic acid and an ephrin receptor tyrosine kinase. Notably, the participation of EphA2 is vital in facilitating the penetration of *C. neoformans* into the CNS. Extensive experimental evidence supports the notion that macropinocytosis is a prominent mechanism facilitating the internalization and transcytosis of *C. neoformans* [24]. *C. neoformans* release various virulence factors that facilitate the spread of infection after it has penetrated the BBB [25]. Glucuronoxylomannan (GXM) and galactoxylomannan (GalXM), key components of the *Cryptococcus* polysaccharide capsule, collectively referred to as glucuronoxylomanogalactan (GXMGal), are recognized as the primary virulence factors of *Cryptococcus* [26,27,28]. *C. neoformans* releases various virulence factors that aid in infection spread by degrading components of the extracellular matrix (ECM) and basement membrane. These activities enable the invasion of tissues and facilitate the spread of the infection [29,30,31,32,33].

## 2. Anatomy and Physiology of the BBB

The brain, as an extremely sensitive environment, requires meticulous regulation of its conditions to maintain functional electrochemical signaling. This signaling is essential for our ability to perceive and react to the external world. To protect against potential harm from pathogens, foreign substances, or unregulated physiological changes, organisms with a CNS have developed a protective barrier known as the BBB. The BBB helps maintain the stability of the CNS and controls the highly selective transport of various substances between the brain and the blood [34,35].

The human BBB is a unique structure essential for neuroscience research and evaluating treatment methods [36]. It is made up of brain microvascular endothelial cells (BMECs), astrocytes, pericytes, neurons, microglia, oligodendrocytes, and neurons [34,37,38,39]. BMECs, also known as brain microvascular endothelial cells, are the primary regulators of the brain’s microenvironment and are a vital component of the BBB. These cells form a protective layer around all microcapillaries in the brain, collectively covering a surface area estimated to range from 12 to 18 square meters in adult humans [40].

Astrocytes, named for their star-shaped morphology, are the most prevalent cell type within the CNS. These glial cells have functions beyond providing support and structure. They are involved in synaptic formation, maturation, plasticity, and neurotransmitter recycling, and make up the neurovascular unit at the BBB. *C. neoformans* must cross this unit to colonize the CNS and cause cryptococcal meningitis (CM). Astrocytes may play a significant regulatory role in the prevention and progression of infection [7,41,42]. They also control the permeability of the BBB [43].

Pericytes are found in most non-epithelial tissues surrounding vessels; however, they are most abundant in the CNS, especially in the retina, where they cover about 30% of the vessel surface with varying frequencies depending on their location [44,45]. Our comprehension of the significance of neurons in maintaining BBB integrity has evolved over the past few decades. However, it is important to note that they do not participate in the formation of the BBB during early brain development [46]. Microglia, commonly known as the brain’s immune cells, actively survey the brain microenvironment, remaining constantly ready to detect antigens through their major histocompatibility complexes [47]. They have important roles in both the adult and developing brain [48]. The phagocytic activity of microglia is crucial for normal brain development as it helps remove defective synapses and contributes to synapse pruning [49].

Oligodendrocytes are a type of macroglia cell that plays a role as neurovascular unit (NVU) cells in the BBB. Oligodendrocyte precursor cells are essential for maintaining the BBB and forming blood vessels throughout life. In return, they receive trophic support [50]. These components work together to create the brain endothelium, which is the primary structure of the BBB (Figure 1).

Brain capillaries have distinct characteristics and functions compared to capillaries found in other tissues and organs: (a) Brain capillaries have fewer or no fenestrations (pores) when compared to non-brain capillaries; (b) The endothelial cells in brain capillaries are tightly interconnected through overlapping and closely joined configurations, facilitated by tunneling junctions, effectively preventing large molecules from passing between endothelial cells [51], effectively preventing large molecules from passing between endothelial cells; (c) Human brain microvascular endothelial cells (HBMECs) exhibit a low endocytosis rate [20,51]; (d) Endothelial cells are surrounded by a continuous basement membrane, and the end feet of perivascular astrocytes surround about 85% of the surface of BMECs outside the basement membrane [20]. Different blood solutes have varying degrees of difficulty when moving from brain capillaries into brain tissue: some pass quickly, others pass slowly, and some cannot pass at all. The brain’s extremely selective permeability implies that the BBB—a barrier that limits solute passage—exists [43]. This structure demonstrates selective permeability, which helps to preserve the physiological homeostasis of the CNS and prevents or minimizes damage to brain tissue caused by toxic substances in blood circulation [39].

The BBB can undergo significant changes as a result of CNS diseases. These changes include the disruption of transcellular junctions, enlargement of the intercellular space, and a significant increase in the barrier’s permeability, allowing large molecules like plasma albumin to pass through [39]. When the BBB is compromised, pathogens can enter the brain, leading to CNS disorders [21]. Additionally, pathogen infections can further compromise the BBB’s integrity and permeability, increasing the likelihood of viruses, immune cells, or cytokines entering the brain. This can worsen CNS inflammation and cause additional brain damage [52]. Research suggests that AIDS patients with cryptococcal meningitis may experience reduced astrocyte activation, which could further compromise the BBB [53].

The BBB is a complex structure located at the interface of the circulatory and central nervous systems. It is made up of BMECs, astrocytes, pericytes, microglia, oligodendrocytes, and neurons and plays a crucial role in maintaining the integrity of the neural microenvironment. Its main function is to regulate the levels of substances in the blood to ensure a stable environment for proper neural function. Additionally, the BBB acts as a protective barrier, preventing toxins and microorganisms from the bloodstream from entering the brain.

## 3. In Vitro Modeling of BBB

In various studies, in vitro models are frequently used to investigate the mechanisms by which microorganisms cross the BBB [12,54]. To assess the effectiveness and safety of medications targeting the CNS and to aid in drug design, it is crucial to have a comprehensive understanding of the penetration mechanisms and physiological responses of the human BBB [34]. This knowledge is vital for the development of appropriate medications and for investigating the underlying mechanisms of various cerebral diseases, particularly those of infectious nature (such as meningitis) caused by pathogens that traverse the BBB. However, due to the intricate multi- and intercellular structures and the limited availability of stable and viable cell sources, there is a scarcity of in vitro models that accurately replicate the architecture and functions of the human BBB despite its significant therapeutic relevance [34].

The in vitro model commonly used to study the BBB involves a 2D Transwell system. In this system, a layer of endothelial cells (ECs) is placed on a porous membrane that separates the upper and lower chambers [55,56,57]. However, this method is unable to fully capture the intricate structures and physiological processes of the human BBB [58,59]. To address the limitations of the 2D BBB model, researchers have turned to organ-on-a-chip devices to create in vivo-like BBB models. These microfluidic chips mimic physiological fluid flow with realistic dimensions and offer technical advantages that enhance BBB modeling [60,61,62]. Significant progress has been made in transitioning from early two-dimensional (2D) static models to the current three-dimensional (3D) dynamic microfluidic chips for in vitro BBB modeling. While the commonly used static in vitro BBB models are relatively simple to construct and allow for convenient measurement of transendothelial electrical resistance, they do not accurately replicate the ideal BBB environment. This is primarily due to their inadequate representation of the correct physiological size and scale, as well as the absence of hemodynamic shear stress, both of which play integral roles in facilitating and sustaining EC differentiation into a specific BBB phenotype [34].

The most basic model for studying the BBB is a single-cell culture model. In this model, BMECs are inoculated onto a membrane that separates two chambers. A culture medium is introduced into the upper and lower chambers of a permeable insert [8]. Researchers have also developed co-culture models, including triple-layer co-culture models (involving BMECs, astrocytes, and pericytes) and double-layer co-culture models (involving BMECs and astrocytes) [8]. Experiments involving the measurement of permeability and drug testing have demonstrated the value of blood–brain barrier-on-a-chip models with endothelium lining the microfluidic device. In these assays, electrodes were integrated into the chips to measure trans-endothelial electrical resistance, and drug contents were measured using high-performance liquid chromatography (HPLC) [60,62,63].

In recent developments, 3D brain or BBB chips have been created to address these challenges using combinations of various cell types (such as ECs, PCs, astrocytes, and neurons) in a 3D collagen or fibrin gel [63,64,65]. However, obtaining human neural cells is challenging; therefore, neuronal components are often omitted or replaced with animal neurons. A recent study presented an integrated organ-on-a-chip consisting of a brain unit with human neurons and a BBB unit containing human brain microvascular ECs, PCs, and astrocytes. This was achieved by recognizing neurons’ significant role in metabolic homeostasis, which is regulated by neurovascular units [66]. While many studies using BBB chips have focused on drug testing, including drug efflux, barrier permeability, and metabolic responses to drugs and inflammatory cytokines, several unexplored aspects of BBB function and interactions require further investigation. Only a few studies have explored the BBB response to brain-infectious pathogens and their behavior in microfluidic chips [67,68].

The interactions between hosts and pathogens during brain infections are complex and dynamic, especially in terms of how the pathogens affect the nervous system. Using a BBB chip with neuronal components to mimic brain infectious diseases would be extremely valuable [8]. Kim et al. created a microfluidic human-neurovascular unit (hNVU) chip to model infectious brain diseases and study how pathogens penetrate the BBB and interact with neural tissues. This chip contains all the necessary cellular and extracellular components and includes a brain unit with astrocytes and neurons derived from human neural stem cells (NSCs) in a 3D hydrogel that mimics the brain’s extracellular matrix (ECM). The hNVU chip allows real-time observation and quantitative assessment of how fungal pathogens penetrate the brain and interact with neural tissues [8]. Researchers used the hNVU chip to screen a wide range of *C. neoformans* mutants, providing a systematic analysis of the mechanisms involved in brain infection by *C. neoformans* and the complex signaling networks at play [69,70]. This approach aims to identify new factors critical for fungal migration across the BBB and neurotropism. Ultimately, these studies could help identify inhibitors that could block the microbial penetration of the BBB, potentially leading to the development of therapeutic agents for treating infectious brain diseases [8].

## 4. Mechanism of Central Nervous System Infection by *C. neoformans*

*C. neoformans* infects the brain, leading to meningoencephalitis, the main clinical manifestation of cryptococcosis. There are three entry sites for neurotropic pathogens: the BBB, the CSF–brain barrier, and the blood–cerebrospinal fluid (BCSFB) [18,19]. Studies suggest that *C. neoformans* has a limited ability to migrate through the choroid plexus to invade the CNS, as the choroid plexus remains normal and free of fungal cells in infected mice [71,72]. Additionally, only a few clinical cases of choroid plexitis have been reported in human patients [73,74,75], indicating that crossing through the BCSFB plays a secondary role in cryptococcal dissemination into the CNS. Instead, clusters of *C. neoormans* have often been observed near the cortical microvasculature of the brain and cerebellum, indicating that *C. neoformans* crosses the BBB for brain invasion [71,72]. To enter the brain, *C. neoformans* must cross the BBB, and three major pathways have been proposed for this: Trojan horse, transcytosis, and paracellular crossing (Figure 2) [29,76].

### 4.1. Trojan Horse Mechanism

The term “Trojan horse mechanism” refers to infections that are taken up by phagocytes, evade being broken down by them, and then enter the brain as the phagocytes pass through the BBB [77]. Recently, this mechanism has become a popular subject of study [56,78,79]. When *C. neoformans* causes a lung infection, it encounters alveolar macrophages in the lungs as one of the first immune responses. These specialized phagocytes quickly engulf cryptococcal cells to destroy them [80]. However, research has shown that macrophages can help spread the infection to the brain through the bloodstream [78,81].

In a mouse model of infection, following intranasal instillation, it was found that alveolar macrophage and dendritic cells are essential for internalizing and transferring *cryptococci* to the lymph node, lungs, and brain [82,83]. This suggests that these cells spread the infection beyond the lungs. This process occurs early during the infection, as shown by detecting fungal loads in mediastinal lymph nodes as early as 24 h after infection [82]. This finding suggests that the dissemination of *C. neoformans* may be an unintended consequence of the antigen-presenting capability of these phagocytes. Remarkably, in transgenic mice where the CD11c+ populations were depleted, the transportation of *C. neoformans* was halted entirely [82]. This provides further evidence for the role of these cells in Trojan horse-like transportation. For *C. neoformans* to use phagocytes as a means of dissemination, it needs to survive within them and then successfully escape after being taken in while also protecting itself from the acidic, oxidative, and nitrosative stresses that macrophages regularly encounter [84]. The expression of inositol phosphosphingolipid-phospholipase C1 (Isc1) is crucial for *C. neoformans* to survive in the acidic pH conditions of mature phagolysosomes [85]. This mechanism leaves extracellular *C. neoformans* vulnerable to the host’s immune responses, leading to local inflammation [86]. However, *C. neoformans* can exit macrophages through a non-destructive process known as vomocytosis, where mature phagosomes containing cryptococcal cells fuse with the cell membrane, releasing the fungi into the external environment [87].

The process of vomocytosis, which involves the expulsion of *C. neoformans* from macrophages, still needs to be completely understood. It is believed that both the host and the pathogen contribute to this process [88]. After the expulsion, both the phagocytes and the fungus coexist, allowing *C. neoformans* to escape from macrophages without causing the typical inflammation associated with cell death [86]. *C. neoformans* produces urease, which aids in vomocytosis [89]. Urease hydrolyzes urea to produce the weak base ammonia [90], increasing the pH of phagosomes and protecting the pathogen from acid-induced damage [89]. This mechanism serves to protect other pathogens from acid-induced damage. However, the elevated pH also induces *C. neoformans* to enter a dormant state, slowing down its replication [89]. Consequently, *C. neoformans* can stay within host cells for a longer time, increasing the likelihood of being transported out of the lungs before the non-lytic expulsion of dormant yeast occurs [29].

The high-affinity FC-γ receptor 3A on phagocytes has been shown to cause cryptococcal meningoencephalitis by facilitating the phagocytosis of cryptococci [91]. MYOC, a cytoskeleton-related gene, is a key regulator of the miRNA–mRNA regulatory network and is significantly downregulated after *C. neoformans* infection [77]. Recent miRNA transcriptomics research has revealed that the cytoskeleton and myocilin, encoded by MYOC, play a role in cryptococcal brain dissemination by modulating the Trojan horse pathway in mice and macaques. This may explain the higher incidence of cryptococcal meningoencephalitis in HIV/AIDS patients, who often have dysfunctional immune cell cytoskeletons [77,92].

A monolayer of human cerebral microvascular endothelial cells (hCMEC/D3) cultured in vitro demonstrated that monocytes infected with *C. neoformans* could cross brain endothelial cells, suggesting that these cells may play a role in neuroinvasion [12,93]. Furthmore, intravital microscopy showed that a significant number of CX3CR1+Ly6Clow monocytes were recruited to the brain microvasculature 12 h after *C. neoformans* infection. These monocytes were observed engulfing *C. neoformans*, adhering to the luminal wall of the brain microvasculature, and transmigrating to the parenchyma [94]. In contrast, the accumulation of CCR2+Ly6Chi monocytes in the brain was observed to begin 14 days after intravenous infection with *C. neoformans*. These monocytes were found to drive brain inflammation in mice and humans [95,96].

Support for the Trojan horse mechanism of brain invasion by *C. neoformans* is evident from clinical studies showing that the efficient internalization of *C. neoformans* by phagocytes is positively correlated with CSF fungal burdens and the risk of death in HIV-associated cryptococcosis patients [77]. A study by Sorrell et al. utilized FITC-labeled cryptococci to investigate their behavior and differentiate between engulfed and un-engulfed cryptococci. Uvitex staining was applied to the latter, and fluorescence microscopy was employed to observe both in vitro BBB crossing and phagocytosis. The results revealed that phagocyte-engulfed cryptococci could migrate across the BBB. The study also found that more C. neoformans than *C. gattii* organisms passed through the BBB using the Trojan horse method [12]. This observation supports the notion that *C. neoformans* is associated with neurological diseases. Another study by Santiago-Tirado et al. in 2017 used live-cell microscopy to provide visual confirmation of phagocytes carrying cryptococci crossing the BBB, thus supporting the initial hypothesis and substantiating the existence of the Trojan horse process [93]. Additionally, the researchers observed that phagocytes traveled from the bloodstream to the brain through a structure known as a “donut hole,” which may impede the movement of *C. neoformans*-containing macrophages along the cerebral vasculature, facilitating the transport of *C. neoformans* [97]. Furthermore, a study by Luo et al. found that the phagocytosis of *C. neoformans* inhibited the chemotaxis of macrophages stimulated by CX3CL1 and CSF-1 [98].

The nuclei of leukocytes play a crucial role in creating gaps and pores between and within endothelial cells. This facilitates the white blood cells to move through endothelial barriers by rapidly restructuring endothelial actin filaments rather than causing the endothelial cells to contract [99]. Further research is needed to establish whether immune cells may traverse the BBB and cerebral blood vessels using a similar mechanism. Most information about the “Trojan horse” mechanism has been obtained indirectly, and the direct evidence provided to date is insufficient. Additionally, most research on mechanisms related to the “Trojan horse” has been conducted using in vitro models, as it is more challenging to observe the outcomes of in vivo studies. The actual process by which cryptococci enter the brain remains to be discovered, as recent research on this mechanism has largely failed to provide evidence that *C. neoformans* traverse the blood–brain barrier in this manner [100].

### 4.2. Transcytosis

*C. neoformans* can penetrate the BBB and infiltrate the central nervous system using a process called transcytosis. This mechanism utilizes the host cell’s endocytic pathways and relies on several host factors [24,84]. Before transcytosis, *C. neoformans* cells adhere to the epithelium, with only a few adhesins identified as mediators of cryptococcal–epithelial interactions [101]. A recent development in the field involves using flow cytometry to quantitatively analyze fungi migration into the brain, confirming that the internalization of *C. neoformans* by brain endothelial cells occurs in vivo [102]. Interestingly, quantitative analysis revealed a higher invasion rate of *C. neoformans* than *C. deneoformans* within brain endothelial cells, indicating the potentially greater virulence of C. neoformans during brain infection in mice [102]. According to histopathological data from Chang et al., cryptococci were initially observed near brain capillaries before appearing in the brain parenchyma, distant from these capillaries, and, after ten days, the presence of cryptococci was detected near the meninges [71]. Cryptococci use various methods, covered in more detail in Figure 3, to facilitate their passage across the BBB.

Cryptococcal hyaluronic acid interacts with the endothelial CD44 receptor, leading to the activation of genes such as CAP1 after the cryptococci utilize host inositol to bind with their inositol transporter. This interaction transactivates CD44, causing the phosphorylation of EPH-EphrinA1 (EphA2), which promotes GTPase-dependent signaling. As a result, the actin cytoskeleton rearranges, cytoskeleton-related proteins are upregulated, and cryptococci are internalized via endocytosis. Once internalized, cryptococcal metalloproteinase Mpr1 interacts with Annexin A2 (AnxA2), facilitating the exocytosis of cryptococci from the endothelial cells and allowing them to cross the blood–brain barrier.

CD44 receptor: Lanser et al., 2023, and Jong et al., 2012, conducted studies that have shown the crucial involvement of CD44 in the adhesion and invasion of *C. neoformans* by brain endothelial cells using an actin-dependent mechanism [24,103]. CD44 acts as both a receptor and an anchor for hyaluronic acid (HA), a large molecule found in the extracellular matrix of blood vessels and the capsule of *C. neoformans*. Chen et al. explained the interaction between CD44 and HA [104]. Importantly, before internalization, C. neoformans specifically bind to CD44 on lipid rafts through hyaluronic acid [105]. According to the findings of Jong et al. from 2008, the study suggests that the standard form of CD44 is expressed on brain endothelial cells, indicating that the preference of *C. neoformans* for the nervous system is not affected by selective binding to a specific CD44 isoform in the brain microvasculature; furthermore, exposure to *C. neoformans* does not lead to increased levels of CD44 [106]. The study by Jong et al. reported that mice lacking CD44 and infected with *C. neoformans* showed a prolonged survival time and a reduced fungal burden in their brains; these observations suggest the potential role of CD44 as a host receptor during the passage of cryptococci across the BBB [103]. The ability of cryptococci to adhere to HBMECs is significantly reduced when the hyaluronic acid on the capsule is removed. Moreover, the expression of the CPS1 gene is closely linked to hyaluronic acid on the surface of the cryptococcal capsule [107]. Two subsequent signaling pathways are activated upon successful attachment of cryptococci to the surface of BMECs. The first pathway involves the CPS1-CD44-PKCα/DYRK3-actin filament signaling pathway, as described by Jong et al. The second pathway is mediated by the CD44-EPH-EphrinA1 (EphA2) tyrosine kinase receptor signaling pathway, as reported by Aaron et al. These signaling pathways induce actin cytoskeleton reorganization, subsequently facilitating phagocytosis [106,108] (Figure 3).

The Ephrin receptor tyrosine kinase, EphA2: In a transcriptomic study conducted by Aaron et al. in 2018, it was found that the expression of ephrinA1, the ligand for the receptor tyrosine kinase EphA2, significantly increased in brain endothelial cells when exposed to C. neoformans. EphA2 is a member of the Ephrin family of receptor tyrosine kinases (RTKs), which is the largest subfamily within the RTK group. Specifically, EphA2 belongs to the Eph A class of RTKs [108,109]. Previous studies have shown that endothelial cells exhibit increased expression of ephrinA1 under various conditions, including ischemia [110], inflammation [111], serum depletion, and higher cell density [112]. The GPI linkage acts as an anchor for the EphrinA1 ligand on the membrane, facilitating its binding and activation of EphA2 receptors. Upon activation, juxtamembrane tyrosine residues on EphA2 receptors undergo oligomerization and autophosphorylation. In an in vitro model simulating the blood–brain barrier, the passage of C. neoformans through the brain endothelium is hindered unless the expression of EphA2 transcripts in brain endothelial cells is silenced or the activity of EphA2 is inhibited using an antibody or inhibitor such as dasatinib. However, when treated with recombinant ephrin A1 or an agonist, *C. neoformans* can traverse the brain endothelium [108]. According to Aaron et al. (2018), the inability of the *CPS1* deletion strain to activate EphA2 with ephrinA1 and induce EphA2 phosphorylation, along with its failure to cross the BBB, suggests the involvement of CD44 in the internalization of *C. neoformans*. These results indicate a potential relationship between these transmembrane proteins expressed by the host and the invasion of *C. neoformans* [108]. The role of EphA2 in cytoskeleton remodeling [113], the initiation of signaling cascades for macropinocytosis [114], and its impact on the integrity of the brain endothelium [115], suggests that *C. neoformans* may utilize EphA2 for central nervous system (CNS) invasion [24]. The specific reasons behind *C. neoformans’* interaction with EphA2 are still under investigation [24]. Additionally, studies conducted by Jong et al., 2008, Kim et al., 2012, and Maruvada et al., 2012, have shown that *C. neoformans* activates various host factors, including PI3K, MAPK, Src kinases, Rac1, and Rho-GTPases, to induce transcytosis across the BBB [116,117,118].

Inositol: The brain contains high levels of a sugar called inositol. When *C. neoformans* acquires inositol, it triggers the upregulation of the cryptococcal CPS1 gene and increases the production of hyaluronic acids. This process promotes the adherence and transcytosis of *C. neoformans*, potentially explaining why *C. neoformans* is prone to infect the brain [119].

Inositol is a highly abundant metabolite in the human brain and has been identified as one of the predominant compounds. Astrocytes, which are located near the blood–brain barrier (BBB), have been observed to possess significant concentrations of inositol. In hypertonic conditions, inositol can be swiftly released from cells [120,121]. The increased occurrence of cryptococcal meningitis can be partially attributed to the elevated levels of inositol present in the brain of the host [122]. According to Liu et al., cryptococci use a complex mechanism to acquire inositol from their surroundings. They also utilize inositol obtained from their hosts, which binds to their inositol transporter and enhances the adhesion of *C. neoformans* to HBMECs [119]. Additionally, it has been documented that increased *CPS1* gene expression precedes the absorption of inositol from the inositol transporter on the cryptococcal cell membrane, thereby inducing the synthesis of hyaluronic acid on the cryptococcal cell membrane (Figure 3). Hyaluronic acid increases the adherence of cryptococci to HBMECs by binding to CD44 receptors on CNS BMECs [119,123].

Metalloprotease Mpr1: *C. neoformans* Mpr1, an extracellular protease, is crucial for establishing fungal infection within the CNS. Mpr1 belongs to the M36 class of fungalysins, which is not yet fully understood and is selectively expressed in certain fungal species [57]. An in vitro model lacking the Mpr1 gene (*mpr1*Δ) showed an inability to breach the endothelial barrier. Additionally, in a mammalian host infected with the *mpr1*Δ null strain, a significant increase in survival was observed, attributed to a decreased fungal burden in the brain and the absence of typical brain pathology associated with cryptococcal disease. In vivo investigations suggest that Mpr1 likely targets the brain endothelium specifically and is not essential for fungal dissemination [57].

Phylogenetic analysis of Mpr1 has unveiled a distinctive pattern that is likely associated with the neurotropic characteristics of *C. neoformans* and the specific role played by Mpr1 in breaching the BBB. This unique function of Mpr1 has the potential to inspire new therapeutic strategies and facilitate brain-specific drug delivery approaches [57,124]. Recent research on the extracellular proteome of *C. neoformans* has identified a secreted metalloprotease called Mpr1 (*C. neoformans* Mpr1). This enzyme has been suggested to be necessary for fungal invasion of the CNS [57,125,126]. According to Vu et al., their findings indicate that Mpr1 enhances the adherence of cryptococci to the surface of endothelial cells, thereby aiding the passage of *C. neoformans* through brain microvascular endothelial cells and into the CNS. This discovery highlights the potential significance of M36 proteases, which are crucial for transcellular traversal in the pathogenesis of fungal infections [57].

Furthermore, transcytosis of cells missing AnnexinA2 and egress through HBMECs is inhibited by one of the 62 protein targets of Mpr1 that was discovered on the surface of the BBB [126] (Figure 3). Furthermore, their study showed that Mpr1 promoted the restructuring of the cytoskeleton in human brain microvascular endothelial cells, thus aiding the penetration of the BBB through the AnxA2-MPR1 interaction [126]. Additionally, in the research by Na Pombejra et al., it was found that one of the 62 protein targets of Mpr1, detected on the surface of the BBB, hinders the transport of cells that lack AnnexinA2 and their passage through HBMECs. The researchers also discovered that Mpr1 encourages the restructuring of the cytoskeleton in HBMECs, enabling the penetration of the BBB by forming a connection with AnxA2-MPR1 [57,126].

### 4.3. Paracellular Crossing Pathway

Apart from traversing epithelial cells, *C. neoformans* can also move between epithelial cells through the disruption of tight junctions, either mechanically or biochemically. Moreover, when the epithelial barriers are compromised or damaged, extracellular *C. neoformans* can readily cross them through a pathway called paracellular crossing or paracytosis [84]. This describes how cryptococci enter the brain through gaps between compromised or injured endothelial cell TJs across the BBB [39,127,128]. Notably, some studies have shown that bioactive or osmotic substances, including histamine, cereport, and bradykinin, can cause transient relaxation of TJs. It is still unclear whether a *Cryptococcus* infection causes these changes and encourages pathogens to breach the BBB through the paracellular route. However, there are numerous studies on *Cryptococcus* breaching the BBB through the paracellular route [129].

The urokinase-plasminogen-plasmin pathway: *C. neoformans* can bind and activate host plasminogen, leading to the conversion of plasminogen to the serine protease plasmin, which is also capable of degrading the BBB [127,130]. Plasmin increases pathogen invasiveness by breaking down the fibrin-enriched extracellular matrix and basement membranes by triggering additional zymogen-protease systems that modify vascular permeability [131]. Research by Stie et al. in 2009 found that the cell wall receptors of cryptococci bind to host plasminogens. When tissue-derived plasminogen activator (tPA) is present, the plasminogen-binding cryptococci induce its conversion into plasmin, which binds and degrades the extracellular matrix and cell membrane, enhancing the cryptococci’s ability to invade the extracellular matrix [127]. Furthermore, co-culturing BMECs with plasminogen-bound *Cryptococcus* increased the ability of the cryptococci to invade the extracellular matrix and facilitated the conversion of plasminogen to plasmin without the need for tPA, indicating that BMECs might be able to activate plasminogen [130]. Additionally, this group discovered that BMECs in co-culture with live cryptococci were stimulated to secrete urokinase, and that urokinase-activated plasminogen on the cryptococcal surface into plasmin, improving the ability of cryptococci to cross the BBB. In contrast, siRNA-mediated silencing of urokinase gene expression or using particular inhibitors of urokinase activity eliminated plasminogen-to-plasmin conversion and decreased the ability of cryptococci to cross the BBB [130]. These results imply that *Cryptococcus* can facilitate its passage across the BBB by activating the human urokinase-plasminogen-plasmin pathway (Figure 4A).

## 5. Virulence Factors Important for Brain Infection

The secretion of multiple virulence factors by *C. neoformans* plays a crucial role in its pathogenesis, facilitating the spread of infection after breaching the BBB and causing brain infections [25]. Research has demonstrated that numerous cryptococcal virulence factors contribute to the extrapulmonary spread of infection; these factors include the capsular polysaccharide, melanin, urease, phospholipase, mating type, phenotypic switching, and mannitol [132]. The morphology of fungal cells and their capsules has been found to influence BBB penetration. During an infection, the production of giant/titan cells is a defense mechanism that protects the cells from being engulfed by macrophages, making it harder for the infection to spread effectively from the lungs to the bloodstream [133]. Various descriptions of *C. neoformans* cell morphological variations have included smooth, mucoid, or wrinkled forms [134].

### 5.1. Capsular Polysaccharide

The capsule of *C. neoformans* (5–7 μm in diameter) is mainly made up of glucuronoxylomannan (GXM) and glucuronoxylomanogalactan (GXMGal) [135]. GXM makes up around 90% of the capsule, while GXMGal constitutes the remaining 10%. During infection, *C. neoformans* releases significant GXM components, which negatively impacts the host immune response [136,137,138]. GXM, as indicated by its name, has a mannose core with variable substitutions of glucuronic acid and xylose, while GXMGal has a galactan core with galactomannan side chains that carry glucuronic acid and xylose. Additionally, the mannose residues in both polymers can be O-acetylated [139]. The polysaccharide capsule is the main determinant of *C. neoformans* virulence; it enables the fungus to evade phagocytosis, suppress humoral and cellular immunity, and protect against oxidative stress [1,140,141,142]. These functions are essential for the successful establishment and progression of infection within the host [143,144,145]. The low immunogenicity of polysaccharide capsules limits the ability of macrophages to engulf fungal cells [146]. Polysaccharide capsules can protect encapsulated cells from macrophage-induced cell death, even after phagocytosis, by defending against reactive oxygen species generated by macrophages [80,147,148].

The phagocytosed *C. neoformans* can reproduce through the Trojan horse mechanism in macrophages and exit the lungs [148]. Subsequently, *C. neoformans* breaches the BBB, leading to the development of meningitis [93]. GXM inhibits dendritic cell activation, antigen presentation, and neutrophil migration as well as neutrophil extracellular traps in human neutrophils. Additionally, GXM can stimulate the production of both pro- and anti-inflammatory cytokines and cause T-cell apoptosis [149,150,151,152]. To reduce the fungal burden in the host, there are two approaches: targeting the destruction of the polysaccharide capsule or inhibiting the release of GXM, which enhances macrophage phagocytosis. In vitro studies have shown that hypo- or acapsular mutants of *C. neoformans* can survive and multiply. However, when tested in a mouse infection model, these mutants exhibit minimal to no pathogenicity. Studies using animal models indicate that the capsular mutants exhibit decreased virulence or avirulence [149,150]. It is crucial to understand capsule biosynthesis to define the biology of *C. neoformans* and potentially discover new therapeutic targets [142].

### 5.2. Melanin

Melanin is a diverse collection of dark pigments found across all biological kingdoms [153]. This polymer consists of covalently bonded indoles, resulting in a granular overall structure. However, the specific structures need to be more well-defined because it is a mixture of polymers with varying pre-indole structures [154]. Melanin is characterized by a negative charge and has been observed in various strains of *C. neoformans*. In encapsulated strains, melanization leads to an increase in cellular negative charge ranging from 3% to 33%, while in an acapsular strain, the charge increased by 86% [155,156]. This suggests that the charge of the cells may impact phagocytosis. The hydrophobicity and charge of the cell surface are known to influence the process of *C. neoformans* phagocytosis [157].

Melanization of *C. neoformans* has been found to impede phagocytosis in vivo [158]. Melanin is considered the second most crucial virulence factor in *C. neoformans*, contributing approximately 14% to the pathogen’s overall virulence [144]. For fungi, melanin serves a dual purpose by providing resistance against radiation and enzymatic degradation in the environment, as well as aiding in the evasion of the human body’s defenses [159]. Melanin can reduce the effectiveness of antifungal treatments and environmental stressors, while also hindering the innate immune system’s ability to clear fungal infections by decreasing the production of reactive oxygen species (ROS) and impairing phagocytosis [160,161,162]. Melanization not only affects the activities of macrophages but also impacts T-cell responses [160,163]. Studies have shown that melanin production by *Cryptococcus* species suppresses the host’s inflammatory and immunological responses [77,164]. In a study conducted by Rosas et al. in 2002, the injection of isolated melanin particles from *C. neoformans* into the peritoneal cavity of animals led to the formation of granulomas in the lung, spleen, and liver. The development of granulomas resembling an immune response to a foreign substance indicates that the immune system may be unable to metabolize the exogenous melanin particles, potentially contributing to this phenomenon [160,165].

Additionally, the melanin influences fluconazole susceptibility and strengthens resistance to Amphotericin B [166]. In the production of melanin, a phenol oxidase compound called laccase (Lac1) is responsible for synthesizing this pigment and integrating it into the cell wall of *Cryptococcus* spp. [167,168]. Lee et al. in 2019 explored the signaling pathways involved in controlling the melanin production in *C. neoformans* [169]. Their findings revealed the crucial role of two kinases, Gsk3 and Kic1, alongside four significant transcription factors, Bzp4, Hob1, Usv101, and Mbs1, in regulating melanin synthesis and other essential steps during the melanization process [169]. Specifically, these core transcription factors responsible for melanin regulation (MRC-TFs) exert specific control over LAC1 and also facilitate connections with the cAMP system, Ace2, and morphogenesis (RAM) regulation, as well as the High Osmolarity Glycerol (HOG) signaling network [170].

### 5.3. Urease

The involvement of the urease enzyme of *C. neoformans* in the virulence of the fungus during infection in humans is well recognized [171]. Stable urease production is necessary for the organism to enter the CNS [172]. The cryptococcal urease, encoded by URE1, plays a crucial role in the invasion of CNS by promoting the sequestration of yeast within microcapillary beds. This mechanism facilitates the transmission of the pathogen from the bloodstream to the brain [173]. Studies conducted on murine infection models of cryptococcosis have indicated that the presence of a urease enzyme has an impact on the traversal of the BBB [174] and the immune responses in the lungs [175].

In a recent investigation, it was shown that urease raises the pH of phagolysosomes, which helps *C. neoformans* survive and adapt within murine macrophages [89,173]. Olszewski et al. (2004) conducted a study to explore the role of cryptococcal urease in the transmission and invasion of cryptococci from the lungs to the CNS. The researchers hypothesized that urease assists in the adherence of *C. neoformans* to brain microvascular endothelial cells during hemorrhagic transmission, thereby promoting the penetration of the BBB [173]. In another study by Charlier et al. (2005), horseradish peroxidase was used to study the entry of cryptococcal yeast cells into the brain parenchyma and the potential damage to the BBB [72]. The researchers observed lesions in the BBB and BMECs, indicating structural and functional changes in the BBB shortly after cryptococcal fungemia. Further investigations by Shi et al. (2010) involved intravenously injecting fluorescently labeled urease-mutant strains of *C. neoformans* into mice and imaging their cerebrovascular systems. The results showed a significant decrease in the ability of the urease mutant strain to penetrate the brain, suggesting that urease may biochemically affect brain endothelial connections [56,174].

Another study discovered that urease activity depended on accessory proteins Ure4, Ure6, and Ure7. Mutations in the core urease protein Ure1, accessory protein Ure7, and transporter protein Nic1 resulted in reduced pathogen invasion into the central nervous system, suggesting a potential impact on the integrity of tight junctions between BBB endothelial cells [176]. Cryptococcal urease is thought to assist in breaking down external urea encountered by the yeast in its natural environment and within the human host. Additionally, endogenous urea produced through the yeast’s metabolic processes could serve as a substrate for urease [177]. By converting urea into ammonia, urease may contribute to the degradation of tight junction proteins and the widening of endothelial cell gaps, leading to compromised tight junctions and increased permeability of BMECs. This facilitates the migration of cryptococci (Figure 4B) [178].

### 5.4. Phospholipase

Phospholipases are important for the pathogen when invading host cell membranes [179]. These enzymes, including phospholipase A, phospholipase B (Plb1), phospholipase C, and phospholipase D [180], play a crucial role in the virulence of *C. neoformans* by promoting capsular enlargement, immunomodulation, and intracellular replication [161]. Plb1, in particular, has been identified as a key virulence factor of *C. neoformans* within the central nervous system (CNS) and contributes to brain injury [161]. It induces capsule enlargement, hampers phagocytosis by macrophages, and is crucial for intracellular replication [181]. Disruption of the *PLB1* gene significantly reduces all three enzyme activities without altering the virulence characteristics of cryptococcal infection. The glycosylphosphatidylinositol anchoring of Plb1 allows it to be strategically positioned in the cell wall, facilitating its rapid release in response to environmental changes [182]. Plb1 plays a vital role in the systemic dissemination of *C. neoformans* from the respiratory system, involving lymph nodes and blood arteries, as well as in the development of pulmonary infection [81]. However, it is not required for transmigration across the BBB and invasion of the brain in mice, particularly within mononuclear phagocytes or through the Trojan horse mechanism [81]. However, it has been demonstrated that *C. neoformans* PLB1 promotes fungal transmigration across an in vitro BBB model.

Furthermore, it activates Rac1, a GTP-binding Rho family protein that regulates the actin cytoskeleton in the host cell [118]. These findings indicate that Plb1 plays a role in the fungal invasion of the brain and central nervous system diseases [161]. Therefore, inhibitors targeting the production and release of Plb1 could be used in treating patients with cerebral cryptococcosis.

For instance, laboratory studies have shown that compounds such as bisquaternary ammonium salts, 1,12 bis-(tributylphosphonium)-dodecane dibromide, and alexidine dihydrochloride effectively hinder PLB1 production and demonstrate anti-cryptococcal activity [183,184].

### 5.5. The Mating Type

Previous studies have shown that in *C. neoformans*, virulence, sexual reproduction, and mating type are closely linked [185,186]. During the mating cycle of *C. neoformans*, when two yeast cells fuse, it leads to the development of hyphae. These hyphae then give rise to branching segments, which further develop into spores and basidia [187]. There are two mating types of *Cryptococcus* spp. in nature, namely, α (*MAT*α) and a (*MAT***a**), with the former being more prevalent in both ambient and clinical isolates [188]. *C. neoformans* exhibits two forms of sexual reproduction: (i) unisexual fruiting, also known as haploid fruiting, involving cells of the same mating type (MATα cells), and (ii) bisexual reproduction, which occurs when two cells possess compatible mating types (i.e., *MAT*α and *MAT***a**) [189]. Sequencing data reveals that the MAT locus in *C. neoformans* is distinctive, spanning over 100 kb and encompassing more than 20 genes [189]. The gene products originating from the MAT locus include various components such as MAP kinase cascade, regulators of sexual reproduction, pheromones, pheromone response, osmosensing, invasive growth, cell wall integrity, and other proteins unrelated to mating [189,190]. The mating type locus of *C. neoformans* differs significantly from mating type loci in other fungi and shares certain characteristics with self-incompatibility systems and sex chromosomes found in animals and plants [189]. During mating, cells release either **a**- or α-pheromones, which attach to G-protein-coupled receptors on the other mating type’s cells [191].

When the mating pheromone binds to its receptor, it activates a G protein, which then triggers the MAP kinase cascade, kicking off the mating process. This process combines haploid yeast cells of different mating types, *MAT***a** and *MAT*α. After this, the α-subunit of the G protein separates. The βγ heterodimers then attract the scaffolds Ste5 and Far1 to the cell membrane [192]. Far1 then binds to several proteins, such as Bem1 and Cdc24, a guanine nucleotide exchange factor for the Rho-type GTPase Cdc42, to facilitate polarized growth towards a mate [193]. Similarly, active Cdc42 and Ste5 recruit proteins like Ste20, Ste11, MEK Ste7, and MAPK Fus3 [194]. Once activated, Fus3 phosphorylates multiple substrates crucial for regulating various cellular processes needed for successful mating. These processes include setting up a transcriptional program, G1 cell cycle arrest, directed polarized growth, and nuclear and cell–cell fusion [192].

The development of *C. neoformans* involves a complex interplay between genes within the α locus and genes located outside of the MAT locus. The small size of spores and desiccated *C. neoformans* facilitates their infiltration into the deeper regions of the alveoli, making the species complexes more pathogenic. Moreover, the interactions between different mating types and their mammalian hosts show variations, as *C. neoformans* α isolates tend to spread into the CNS more than congenic a-type isolates [186]. Moreover, hybridization offers an extra selective advantage to *C. neoformans* by promoting the growth of hyphal forms during mating, which in turn enhances their ability to search for nutrients over a larger surface area [186].

### 5.6. Phenotypic Switching

Pathogens often change their shape by adjusting the expression of specific genes in response to the host environment. These changes are known as “virulence” factors linked to the pathogen’s ability to adapt and survive in the host. When *C. neoformans* enters the host environment, it undergoes changes related to virulence, such as alterations to the cell surface and size. Surface changes include processes like melanin production, modifications to the cell wall composition, and the development of a polysaccharide capsule. Recent studies have shown that changes in cell size are important virulence traits in *C. neoformans*.

The common yeast cell, the massive titan cell, and the small micro and titanide cell are examples of distinct cell size phenotypes. Changes in cell surface and size affect the interaction between the host and the pathogen. However, the impact of genetic polymorphisms on these traits remains unclear [195]. α-glucan, β-glucan, melanin, chitin, and chitosan are all found in the cell walls of normal cells. Titan cells (diameter > 10 μm) have a thicker cell wall with a layer of mannan, less glucan, and more chitin [196]. Thickened cell walls are another feature of microcells (diameter less than 1 μm) [197]. Titanides, with an oval form and a diameter ranging from 2–4 μm, have thinner cell walls than typical cells [135]. For a long time, it has been recognized that changes in the cell surface are crucial for determining the virulence of *C. neoformans*. Since mammalian cells lack cell walls, the immune system detects various fungal cell wall components, such as β-glucan and chitin, to initiate an immune response against fungal pathogens [195]. *C. neoformans* possesses a unique morphotype known as a titan cell, with a diameter exceeding 30 µm (including the polysaccharide capsule). Interestingly, these large, polyploid titan cells give rise to aneuploid cells of normal size during infection. In some cases, the aneuploidy observed in the progeny cells often contributes to their survival within the host, especially in challenging conditions such as oxidative stress and antifungal treatment. The larger size of titan cells helps them avoid being engulfed by macrophages in the lungs, but it also limits their ability to pass through biological barriers and subsequently disseminate to the brain [198].

### 5.7. Mannitol

*C. neoformans* can produce D-mannitol in both culture and infected animals [199]. D-mannitol is a reliable indicator for evaluating the severity of experimental cryptococcal meningitis. Furthermore, the production of D-mannitol by *C. neoformans* can impede the ability of phagocytes to eliminate the infection and can lead to brain edema by scavenging reactive oxygen intermediates (ROIs) within the host [200,201]. The presence of mannitol in the cerebrospinal fluid of patients with CM has the potential to exacerbate increased intracranial pressure and contribute to neurological harm [202].

## 6. Conclusions

*C. neoformans* was identified as a pathogen in the late 1800s. It is important to understand how cryptococci interact with the BBB because this interaction is crucial in the development of cryptococcal meningitis. Additionally, due to the connection between the sinuses and the cerebral cavity, *Cryptococcus* can easily penetrate the brain. This review focuses on the various routes through which *Cryptococcus* enters the brain and the process it undergoes to traverse the BBB. The human body is a complex system with multiple layers; therefore, there are still many processes that need to be investigated. Three mechanisms related to cryptococci crossing the BBB have been reported so far. We hypothesize that the methods by which *C. neoformans* traverse the BBB may differ, which could affect the speed and efficiency of their entry. Furthermore, immunocompromised patients are at a higher risk of contracting *C. neoformans* due to the lack of immunocytes, which impairs the clearance of the pathogen and how HIV facilitates *C. neoformans* penetration of the BBB. Future research should focus on determining whether *C. neoformans* penetrate the BBB through all three mechanisms or only one or two of them, resolving differences in the quantity or rate at which *C. neoformans* enter the brain via the same mechanism, and determining which mechanism is more likely to cause meningitis. Moreover, while the discovery of transcellular traversal was significant, it is still uncertain which of the three processes for crossing the BBB is the most dominant. The process by which cryptococcal cells traverse the BBB has been clarified, which will aid in developing appropriate medications to prevent them from entering the brain. This preventative measure will substantially reduce the incidence and mortality of cryptococcal infections in the central nervous system. Over the past 40 years, the virulence mechanisms of *C. neoformans* have been extensively researched. Despite the scientific community’s considerable efforts, controlling and mitigating its effects still poses significant challenges. One of these challenges is fully characterizing the capsule, which remains the most researched structure of this yeast. Nonetheless, the specific synthesis procedure is still a mystery. Other virulence features, such as intracellular pathogenesis—a unique characteristic of *Cryptococcus*—also have similar knowledge gaps, and it is unclear how much this trait contributes to the overall virulence picture. Therefore, further research is necessary to fully comprehend the pathogenic mechanisms of this fungal pathogen. In conclusion, preventing cryptococcal entry into the brain may facilitate the development of a novel therapeutic approach to aid individuals with cryptococcal meningitis.

## Figures and Tables

**Figure 1 jof-10-00586-f001:**
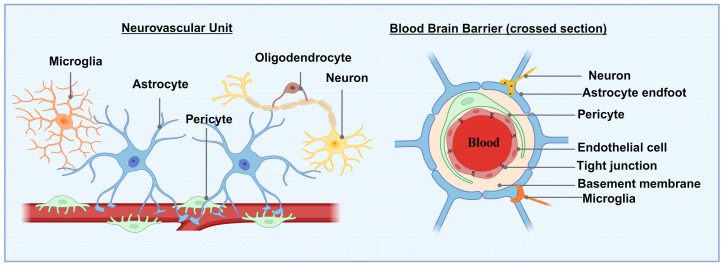
Schematic illustration of the BBB.

**Figure 2 jof-10-00586-f002:**
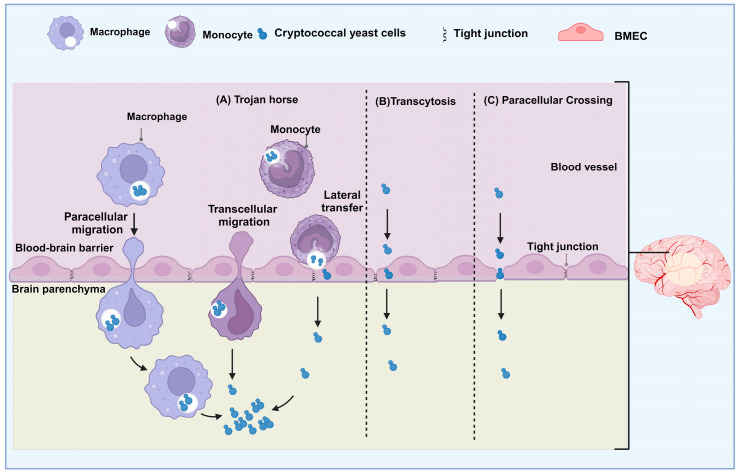
The illustration of the mechanisms by which cryptococcal yeast cells cause brain infection. *C. neoformans* can cross the BBB either indirectly through a Trojan horse mechanism or directly through transcytosis and paracellular crossing. (A) The “Trojan horse” mechanism involves phagocytes carrying ingested *C. neoformans* across the BBB through paracellular and transcellular migration. Phagocytes containing C. neoformans can also directly transfer the fungal cell to brain endothelial cells through lateral transfer, leading to BBB crossing of the organism. (B) Transcytosis mechanism, *C. neoformans* across the barrier through direct endocytosis of brain microvascular endothelial cells without disrupting the intercellular tight junction. (C) Paracellular crossing, *C. neoformans* penetrate between barrier cells through loosened tight junctions and may or may not lead to tight-junction disruption.

**Figure 3 jof-10-00586-f003:**
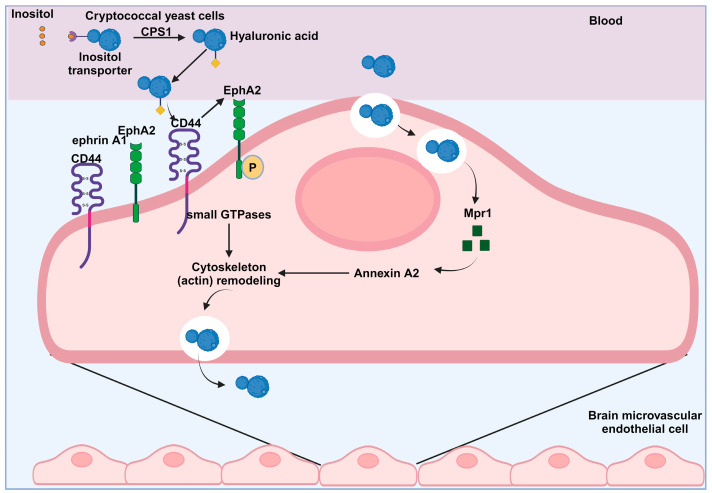
The process by which Cryptococci traverse the BBB through transcytosis. The process by which Cryptococci traverse the BBB through transcytosis. Cryptococcal hyaluronic acid interacts with the endothelial CD44 receptor, leading to the activation of genes such as CAP1 after the cryptococci utilize host inositol to bind with their inositol transporter. This interaction transactivates CD44, causing the phosphorylation of EPH-EphrinA1 (EphA2), which promotes GTPase-dependent signaling. As a result, the actin cytoskeleton rearranges, cytoskeleton-related proteins are upregulated, and cryptococci are internalized via endocytosis. Once internalized, cryptococcal metalloproteinase Mpr1 interacts with Annexin A2 (AnxA2), facilitating the exocytosis of cryptococci from the endothelial cells and allowing them to cross the blood-brain barrier.

**Figure 4 jof-10-00586-f004:**
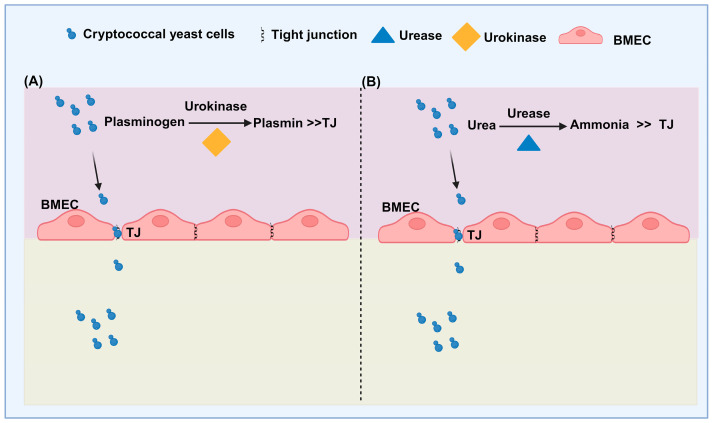
The roles of urokinase and urase in paracellular crossing through BBB. (**A**) The activation of *Cryptococcus* caused HBMEC to release urokinase. On the surface of cryptococci, urokinase stimulates the conversion of host plasminogen into plasmin. Plasmin degrades the extracellular matrix (ECM), facilitating *Cryptococcus* crossing of the BBB. (**B**) The urease enzyme secreted by *Cryptococcus* converts urea into ammonia. The accumulation of ammonia produced by urease can potentially damage cellular junction proteins, thereby promoting the entry of cryptococcal yeast cells into the brain.

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
