# Peer review of "Mechanisms and Virulence Factors of Cryptococcus neoformans Dissemination to the Central Nervous System"

_jof, 2024, doi:10.3390/jof10080586_

Round 1

Reviewer 1 Report

The review submitted by Ammar Mutahar Al-Huthaif and colleagues is very well written and focuses on a topic of great importance for public health.

Humans are exposed to fungi at all times. Most individuals are immune to this aggression and do not develop the disease; however, there is a group that develops the disease. As expected, in a situation of weakness or immunosuppression, humans are susceptible to fungal infections. These microorganisms have different pathogenicity mechanisms and can easily evade the host's immune response and consequently successfully establish the infection.

However, I missed some classic articles that can be added and enrich the manuscript, they are:

a) Immunomodulatory Role of Capsular Polysaccharides Constituents of Cryptococcus neoformans. Front Med (Lausanne). 2019 Jun 19;6:129. doi: 10.3389/fmed.2019.00129 (this review describes immunomodulatory mechanisms triggered by the polysaccharide components of the capsule)

b) Capsular polysaccharides galactoxylomannan and glucuronoxylomannan from Cryptococcus neoformans induce macrophage apoptosis mediated by Fas ligand. Cell Microbiol. 2008 Jun;10(6):1274-85. doi: 10.1111/j.1462-5822.2008.01125.x. (This manuscript describes the proapoptotic role of capsular polysaccharides, inducing macrophage apoptosis)

c) Cryptococcus neoformans glucuronoxylomannan induces macrophage apoptosis mediated by nitric oxide in a caspase-independent pathway  . Int Immunol. 2008 Dec;20(12):1527-41. doi: 10.1093/intimm/dxn112. (This manuscript describes the proapoptotic role of capsular polysaccharides, inducing macrophage apoptosis)

d) Capsular polysaccharides from Cryptococcus neoformans modulate production of neutrophil extracellular traps (NETs) by human neutrophils. Sci Rep. 2015 Jan 26;5:8008. doi: 10.1038/srep08008 (This manuscript describes the mechanism of negative modulation of the GXM polysaccharide on the induction of NETs)

e) Inhibition of Microbicidal Activity of Canine Macrophages DH82 Cell Line by Capsular Polysaccharides from Cryptococcus neoformans. J Fungi (Basel). 2024 May 8;10(5):339. doi: 10.3390/jof10050339. (This study describes the pathways and mechanisms involved in the macrophage apoptosis promoted by GXM through NO generations)

It would be important to add these classic manuscripts to enrich the content of this important review.

The review submitted by Ammar Mutahar Al-Huthaif and colleagues is very well written and focuses on a topic of great importance for public health.

Humans are exposed to fungi at all times. Most individuals are immune to this aggression and do not develop the disease; however, there is a group that develops the disease. As expected, in a situation of weakness or immunosuppression, humans are susceptible to fungal infections. These microorganisms have different pathogenicity mechanisms and can easily evade the host's immune response and consequently successfully establish the infection.

However, I missed some classic articles that can be added and enrich the manuscript, they are:

a) Immunomodulatory Role of Capsular Polysaccharides Constituents of Cryptococcus neoformans. Front Med (Lausanne). 2019 Jun 19;6:129. doi: 10.3389/fmed.2019.00129 (this review describes immunomodulatory mechanisms triggered by the polysaccharide components of the capsule)

b) Capsular polysaccharides galactoxylomannan and glucuronoxylomannan from Cryptococcus neoformans induce macrophage apoptosis mediated by Fas ligand. Cell Microbiol. 2008 Jun;10(6):1274-85. doi: 10.1111/j.1462-5822.2008.01125.x. (This manuscript describes the proapoptotic role of capsular polysaccharides, inducing macrophage apoptosis)

c) Cryptococcus neoformans glucuronoxylomannan induces macrophage apoptosis mediated by nitric oxide in a caspase-independent pathway. 2008 Dec;20(12):1527-41. doi: 10.1093/intimm/dxn112. (This manuscript describes the proapoptotic role of capsular polysaccharides, inducing macrophage apoptosis)

d) Capsular polysaccharides from Cryptococcus neoformans modulate production of neutrophil extracellular traps (NETs) by human neutrophils. Sci Rep. 2015 Jan 26;5:8008. doi: 10.1038/srep08008 (This manuscript describes the mechanism of negative modulation of the GXM polysaccharide on the induction of NETs)0

e) Inhibition of Microbicidal Activity of Canine Macrophages DH82 Cell Line by Capsular Polysaccharides from Cryptococcus neoformans. J Fungi (Basel). 2024 May 8;10(5):339. doi: 10.3390/jof10050339. (This study describes the pathways and mechanisms involved in the macrophage apoptosis promoted by GXM through NO generations)

It would be important to add these classic manuscripts to enrich the content of this important review.

Reviewer 2 Report

This is a thorough, if uncritical, review of virulence factors in the Cryptococcus neoformans and Cryptococcus gattii species complexes, which are referred to in this paper as. Cryptococcus neoformans, using the older terminology. A good understanding of the molecular biology is exhibited but a lesser understanding of the pathobiology. The fact that melanin, hyaluronic acid and titan cells have not been adequately demonstrated to occur  in infected humans does not appear to cause the skepticism in the authors that I have, but I respect the authors’ right to believe whatever they want. The first part of the review has a little more paraphrasing and obvious material than I would prefer, suggesting it was written by a different author than later in the manuscript. For instance , lines 143-148 paraphrase lines 84-91 and lines 176-179 paraphrase earlier material. Lines 46-47 are incorrect about predisposing factor to infection.  Reference to “organisms with a central nervous system” on 82-83 is obtuse. Referring to macrophages being introduced “through the nose”  on line 247 is strange.  The statement that “C. neoformans is more likely to be associated with neurologic diseases” (lines 306-307) is an odd statement about a pathogen which is a major cause of meningoencephalitis. I do not know what is meant by a “dysfunctional immune cell cytoskeleton” (line285) .

There are other current reviews of the virulence factors in organisms within the Cryptococus neoformans and Cryptococcus gattii species complexes  (PMID  33497839, 25256589, 31119976, 36294634,  and 32956032) but this is also a current , useful review, with an emphasis on the passage of organisms from blood to brain.

see above
